# Few-shot graph link prediction with domain adaptation

## Abstract

Real-world link prediction problems often deal with data from multiple domains, where data may be highly skewed and imbalanced. Similar problems in computer vision are often referred to as Few-Shot Learning (FSL) problems. However, for graph link prediction, this problem has rarely been addressed and explored. In this work, we propose an adversarial training-based framework that aims at improving link prediction for highly skewed and imbalanced graphs. We introduce a domain discriminator on pairs of graph-level embedding. We then use the discriminator to improve the model in an adversarial way, such that the graph embeddings generated by the model are domain agnostic. We test our proposal on three benchmark datasets. Our results demonstrate that when domain differences exist, our method creates better graph embeddings that are more evenly distributed across domains and generate better prediction outcomes.

## 1 Introduction

Real-world link prediction problems often deal with graphs with nodes from various domains. For example, we can separate the items from a product purchase network into multiple domains such as books, electronics, luxury items. A financial transaction network can have data coming from various countries. Different domains may have different distributions, but more importantly, domains are often not balanced in most real-world datasets. For example, we may expect to see hundreds of thousands of nodes for books but only a few hundred for luxury items. There might also be billions of financial transactions in developed countries but much fewer in developing countries. We observe a similar phenomenon even in academic graph datasets. For example, in the ogbn-arxiv (Hu et al. (2020)) dataset, which includes pre-prints from 40 computer science subjects, papers from the most popular subject take up to 16.13% of all the nodes. In contrast, the least represented subject only has 0.02% of the data.

Training graph models with imbalanced data without precaution significantly downgrade model performances, especially for domains with fewer samples. The training process is overwhelmed by the popular domains that have abundant data. These popular domains act as noise, hampering the model to fit for small domains effectively. Despite the prevalence of this problem, current research on this topic is not adequate (Zhao et al. (2021); Shi et al. (2020)).

To improve model performance for small domains, we propose to train domain agnostic embedding that we can use for downstream tasks. The intuition is that if the latent representation learned by the graph model does not contain any domain-specific information, and therefore is domain-agnostic, it becomes more robust to domains with fewer data and even novel domains. We propose to use adversarial learning and force the graph model to learn domain agnostic embeddings.

Our idea of building domain-agnostic features across domains aligns with the few-shot learning (FSL) problem in computer vision (CV). FSL (Fei-Fei et al. (2006); Fink (2005)) refers to the type of ML algorithm that works with limited labeled data. A common FSL technique is to learn domain-agnostic features first (with/without external data) and then let the model converge fast on to domains with limited shots (Long & Wang (2015)). It is relatively easier for CV tasks because stacked convolutional neural networks used in CV usually learn features in a general-to-specific sequence. In contrast, as stacking graph neural networks (GNN) layers only explore larger neighborhoods, the same method couldn't be applied directly to graph data.

In addition, the CV literature has explored the idea of using adversarial learning to build better domain agnostic features. Domain-Adversarial Neural Network (DANN) (Ganin et al. (2016)) first introduces the idea of using a discriminator to align and separate semantic probability distributions. Further research (Motiian et al. (2017)) shows that it helps reduce the number of samples needed for each domain/class. Essentially, our work follows this path but further improves it with the enclosing subgraph idea to achieve better performances for graph link predictions.

We find that the method of extracting enclosing subgraphs, proposed in the SEAL model Zhang & Chen (2018) suits well for our setting because it not only boosts the number of trainable samples but also gives them different topological features. It is similar to what "cropping" does for images. CV Research has shown that data augmentation method, such as cropping (Qi et al. (2018); Zhang et al. (2018b)), introduces invariances for models to capture and is effective for FSL problems.

We summarize our main contributions as follows:

- We propose the idea of learning domain-agnostic graph embedding to improve the quality of link prediction by using adversarial learning and enclosing subgraphs.
- We define the concepts of "shots" for graph data, making it significantly easier to train, describe, and analyze graphs with multiple imbalanced domains.
- We use t-SNE plots to demonstrate the effectiveness of our method. It also allows practitioners to decide the expected gain from our proposed ideas.

## 2 RELATED WORKS

### 2.1 LINK PREDICTION

Link prediction focuses on predicting future or missing links between a pair of nodes. The current state-of-the-arts link prediction method is SEAL (Zhang & Chen (2018)). The core idea of SEAL is to extract *enclosing subgraphs* for a set of sampled positive and negative links. To be more specific, a positive link is a pair of nodes connected with an edge, and a negative link is a pair of nodes that are not connected. An enclosing subgraph is the union set of $h$-hop neighboring nodes of the pair of nodes. This data-preprocessing step lets SEAL transform the link prediction problem to a (sub)graph classification problem. It uses graph neural networks (GNNs) to generate the graph embedding, which feeds into a classifier and generates new predictions. Other than SEAL, Graph Auto-Encoder (GAE, Kipf & Welling (2016)) is an alternative approach. GAE works by generating node embedding for all nodes and then model the link likelihood based on the concatenated node embedding vectors for any node pairs. However, recent researches have shown this method heavily relies on the assumption that similar nodes have similar connections. In addition to these GNN-based methods, traditional heuristic methods, such as common neighbor (CN) and Adamic-Adar (AA), provide a completely different alternative. Given a pair of nodes, these non-parametric methods describe the topological structure of the surrounding neighborhood with a score. The advantage is that these methods could be used directly without training, but it's challenging to utilize node/edge features with these methods.

### 2.2 FEW-SHOT LEARNING AND DOMAIN ADAPTATION

Few-shot learning is a special type of machine learning problem, which aims at obtaining good learning performances when supervised evidence information is limited (Wang et al. (2020)). Common FSL methods include data augmentation (cropping, rotating, generative models, etc) and meta-learning (Sun et al. (2019)). Domain Adaptation is a type of Transfer Learning (TL) technique. It focuses on transferring learned knowledge from one domain to another while maintaining good performances. The core idea of domain adaptation is to learn a domain invariant representation, which is usually done by maximizing mutual information (Hjelm et al. (2019)) or training in an adversarial network (Goodfellow et al. (2014)).

Some recent works focus on the problem of few-shot learning with domain adaptation. The work from Motiian et al. (2017) also uses adversarial networks as the solution, but it introduced a 4-class confusion matrix classifier as the discriminator. The work from Zhao et al. (2020) explicitly added source/target per-class separation before doing embedding feature learning with domain adaptation.

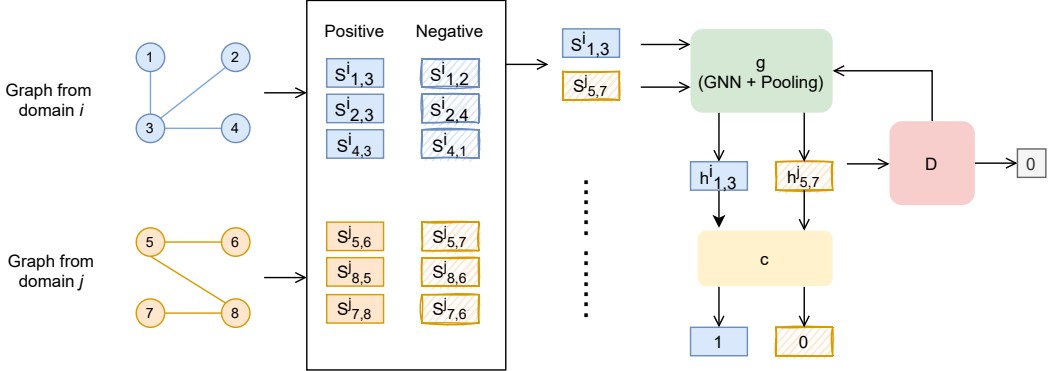

Figure 1: **Few-shot Link Prediction with Domain Adaptation.** Colors denote different domains. The positive/negative outcome is denoted by texture. Solid boxes are positive links, and slash-textured boxes are negative links. Similar to SEAL, we start by extracting enclosing subgraphs around positive/negative links in graphs. A pair of enclosing subgraphs are sent to model $g$, composed of a Graph Neural Network and a graph pooling layer, to generate a pair of graph embeddings. A Discriminator ($D$) is set up on the side to determine whether this pair of graph embeddings come from the same domain. The pair of graph embeddings also enter in model $c$ to classify link existence.

## 2.3 ADVERSARIAL LEARNING

To learn domain agnostic models, we need to ensure that the learned graph embedding does not contain domain-specific information. To achieve this, a commonly used technique is adversarial learning, in which two models are trained against each other and play a minimax game based on game theory. For example, in the popular Generative Adversarial Network (GAN) (Goodfellow et al. (2014)), adversarial learning helps generators build more realistic images by having a discriminator that identifies whether an image is synthetic or real. There have been prior works done on using adversarial training on graph-related tasks. The work from Wang et al. (2018) uses GAN to improve various graph-related tasks, such as link prediction and node property prediction, by generating "fake" graphs. The work from Lei et al. (2019) also uses GAN to solve the challenging temporal link prediction problem for weighted dynamic networks.

## 3 METHODOLOGY

### 3.1 PROBLEM DEFINITION

We define the few-shot link prediction problem with imbalanced domains as follows: in a given graph space $\mathbb{S}$, we have access to graphs from $m$ ($m \in \mathbb{N}, m \geq 2$) number of domains. Different domains may have variations in their marginal distributions. Among the $m$ domains, $m-1$ ($k \in \mathbb{N}^+$) domains have $k$ number of small graphs, which we define as graph "shots". The last domain only has 1 shot. Our objective is to learn a link prediction model that leverages the domain differences to improve the performance in the domain with 1 shot of data.

### 3.2 NOTATIONS

In terms of graph-related notations, each graph is denoted by $G = (V, E, d)$, where $V$ is the set of vertices, $E$ is the set of edges and $d$ is the type of domain, which could be a one-hot encoded vector with one extra digit representing any unseen domain. In the adjacency matrix $A$ of $G$, $A_{u,v} = 1$ if edge $(u, v) \in E$, and otherwise $A_{u,v} = 0$. For any node $u$, we use $\mathcal{N}(u)$ to denote the set of neighboring nodes of $u$. In addition, we use superscript as a shorthand to denote the domain class. For example, for a graph from domain $i$, we would denote it as $G^i$.

### 3.3 DOMAINS

For our setting, we have a total of $m$ domains and each domain has one or more graphs or subgraphs $D^k = \{G_1^k, ...\}_{k=1}^m$, where $D^k$ denotes the $k^{th}$ domain and graph $G_1^k$ is a realization of random variable $\mathcal{G}^k$. The effective of our method compared to baseline is based on the assumption that the distribution of the random variable $\mathcal{G}$ is different across different domains $p(\mathcal{G}^k) \neq p(\mathcal{G}^{k'})$ if $k' \neq k$.

### 3.4 LINK PREDICTION

Here is the general form of any link prediction models: given $(u, v, G)$, where $u$ and $v$ are a pair of nodes in graph $G$, the likelihood of having a link between $u$ and $v$, $y_{u,v}$, is modeled by function $f : (u, v, G) \rightarrow y_{u,v}$. This function could be trained with a union set of positive and negative samples. Positive samples are sampled from existing edges while negative samples are built with node pairs with $u$ and $v'$, $v' \notin \mathcal{N}(u)$.

Based on the ideas from the SEAL framework, the first step is to extract an $h$-hop enclosing subgraph around edge $(u, v)$. We denote this subgraph as $S_{u,v}$ and the process of generating the enclosing subgraph as $s(u, v, G) = S_{u,v}$. Function $f$ could be rewritten as $f : S_{u,v} \rightarrow y_{u,v}$.

In general, $f$ is composed with two separated functions $g$ and $c$ and could be written as $f = g \circ c$. Function $g$ generates the graph embedding vector $g : S_{u,v} \rightarrow h_{S_{u,v}}$. Function $c$ takes in the graph embedding and generates the prediction $\hat{y_{u,v}}$.

Here we have the supervised loss function.

$$\mathcal{L}_{cls}(g, c) = \mathbb{E}[\ell(c(g(S_{u,v})), y_{u,v})]$$
$$= -\frac{1}{N} \sum y_{u,v} \cdot \log(c(g(S_{u,v}))) + (1 - y_{u,v}) \cdot \log(1 - c(g(S_{u,v})))$$

, where $\mathbb{E}$ is the statistical expectation of any loss function $\ell$. In this case, $\ell$ is the standard binary cross-entropy loss.

The input of the discriminator $D$ is a pair of graph embedding $h^i$ and $h^j$ generated by $g$. As stated above, here we use superscript to denote the domain information. The objective is to determine whether these two subgraphs are from the same domain. Since we know from which graph $G$ these subgraphs are generated, and we know the domain information of each $G$, we can easily get the ground truth $d^{i,j}$. The loss function here could be binary cross-entropy as well. An alternative design is to let the discriminator predict the exact domain that the graph embedding is from. We select the current one because it is easier to apply it to novel domains.

$$\mathcal{L}_{dis}(D) = \mathbb{E}[\ell(D(g(S_{u,v}^i), g(S_{u',v'}^j)), d^{i,j})]$$

Given the two loss functions above, we could get the total loss for the input of a pair of subgraphs from domain $i$ and $j$. Since our objective is to make sure the graph embedding does not contain domain-specific information, we would like to minimize the supervised training loss while maximizing the discriminator's loss.

$$\mathcal{L}_{total}(g, c) = \mathbb{E}[\sum_{i=1}^m \ell(c(g(S_{u,v}^i)), y_{u,v}^i) - \alpha \cdot \sum_{i=1}^m \sum_{j=1}^m \ell(D(g(S_{u,v}^i), g(S_{u',v'}^j)), d^{i,j})],$$

where $\alpha$ is hyperparameter to adjust the weight of the discrimination term compared to the supervised loss.

## 4 EXPERIMENTS

### 4.1 DATASETS

In this study, we compared the model performances using three different datasets, including ogbn-products, ogbg-ppa and the protein-protein interaction (PPI) dataset.

Both the ogbn-products and ogbg-ppa dataset comes from the Stanford Open Graph Benchmark (Hu et al. (2020)). There is a specific category of benchmark dataset on OGB for link prediction. However, we could not use those link-prediction benchmark datasets as they do not contain domain information. Therefore, we select ogbn-products and ogbg-ppa from the node/graph property prediction task and convert them into link prediction tasks. The protein-protein interaction (PPI) dataset comes from the GNN benchmark dataset (Dwivedi et al. (2020)) and originally published together with the GraphSage paper(Hamilton et al. (2018)). Its original purpose is also node property prediction and we re-purpose it as a link-prediction problem.

**ogbn-products** The ogbn-products dataset is a single graph with 2,449,029 nodes and 61,859,140 edges, representing the Amazon product co-purchasing network in 47 categories. Node feature is a 100-dimension word embedding vector extracted from product description. During the data preprocessing, we select 23 domains with more than 10,000 nodes and split the nodes into 23 subgraphs, each of which represents a single domain. Then, within each domain, we randomly selected 40% of the nodes as the test data and used the remaining 60% to generate training "shots". To ensure the robustness of our results, we run the random sampling 10 times to generate 10 sets of independent experiment data. In each experiment, within each domain, we start with a random node, run Breadth-first search (BFS) until the cumulated amount of nodes exceed 5% of all the nodes in the training graph (in other words, 3% in the original domain subgraph). All the nodes visited by BFS are considered as a "shot". Then we start from another random node from the rest of the graph, repeat the BFS process until a total of 10 shots are generated for each domain. To build the few-shot learning with an imbalanced domain setting, for 22 out of 23 domains, we arrange all 10 shots to the training set and for the last domain, we arrange 1 shot to the training set. The training set is then randomized and fed into the model. The evaluation is done for the imbalanced domain only.

**ogbg-ppa** The ogbg-ppa dataset include 158,100 graphs representing the protein-protein association network in 1581 different species that cover 37 broad taxonomic groups. The average number of nodes is 243.4 and the average number of edges is 2,266.1. Edges in this dataset are associated with a 7-dimensional feature vector . We chose to pick broad taxonomic groups as our criteria to separate domains. We select 50 graphs from each domain as our test set. For our training set, we selected 10 graphs from each domain to build the data for one experiment and we created 10 independent experiments to keep our result robust. Similar with what we do for ogbn-products, we put 1-shot of data for the evaluation domain into the training set and 10-shot for all the other domains. Note that the graphs in ogbg-ppa do not have node feature data. Instead, only edge feature data is available.

**PPI** The PPI dataset is another protein-related dataset. It includes 24 graphs representing the human protein-protein interaction network in 24 types of human tissues. The average number of nodes is 2372.7 and the average number of edges is 68508.7. Each node has 50 features. We follow the same procedure as we use for the ogbn-products dataset to split test and train, generate shots and produce experiment data.

## 4.2 BASELINE METHODS

We compared the performances of our proposed methods with several commonly used link-prediction methods, including SEAL, Graph AutoEncoder (GAE) and heuristic methods including Common Neighbor (CN), Adamic-Adar (AA) and Personalized PageRank (PPR).

**SEAL** SEAL is the current state-of-the-art method for link prediction and provides a lot of inspiration to this work. The core idea is to create enclosing subgraphs around a selection of positive and negative node pairs. The distance between the target pair of nodes and all the other nodes is calculated within each subgraph. Then, SEAL uses a GNN to compute graph-level embedding for each subgraph and use the graph embedding to predict the probability of having a link between each node pair.

**GAE** GAE employs a standard GCN to generate updated node embedding. For each positive and negative node pair, the node embedding vectors for the pair of nodes are concatenated and then used to predict the probability of having a link.

**CN** As a heuristic method, there is no training process needed for CN. The common neighbor method computes the total number of common neighbors between any node pairs and uses the score to predict the probability of having a link. The mathematical formula for CN is usually denoted by

$$CN = |\mathcal{N}(x) \bigcap \mathcal{N}(y)|$$

**AA** Adamic-Adar is another heuristic method. Unlike CN, instead of looking into the first-order neighbor of a node pair, AA pays attention to the second-order neighbor of the node pair. Its mathematical formula could be denoted by

$$AA = \sum_{z \in \mathcal{N}(x) \bigcap \mathcal{N}(y)} \frac{1}{\log(|\mathcal{N}(z)|)}$$

**PPR** Personalize PageRank computes the stationary distribution of a random walker starting at $x$ and ending at $y$ with restarts. Here since we are focusing on bidirectional graphs, we need to compute the scores for both from x to y and from y to x. The mathematical formula is denoted by the following equation, where $q$ stands for the stationary distribution.

$$PPR = q_{xy} + q_{yx}$$

## 4.3 Experiment Configuration

All the heuristic methods (CN, AA & PPR) can be used directly without training. To ensure the evaluation is fair, we tested all six methods on the same set of node pairs for evaluation.

All three NN-based methods (our method, SEAL & GAE) requires training on a selection of positive and negative node pairs. To ensure a fair comparison, we trained all these three methods on the same sets of node pairs. As mentioned before, we generated 10 sets of experiment data for training. The final results for these NN-based methods are average performances in all 10 experiments. In addition, to reduce the size of training data, we sample a proportion of positive node pairs and then sample the same amount of negative node pairs for each dataset. The exact proportions for each dataset is consistent across all three NN-based methods and is in-fact determined based on the number of edges and the size of features.

In all three GNN-based methods, we use a three layers GCN with 32 neurons on each layer. Classic GCN models only take node features, and it is what we use for ogbn-products and PPI. However, as mentioned earlier, the ogbg-ppa dataset only has edge features. To utilize edge features, we modified the message passing rule such that the message includes the concatenated node feature, which in this case is the calculated distance value between each node and the target node pair, and edge features. In our method and SEAL, where a graph level pooling is needed, we use the pooling method proposed in DGCNN (Zhang et al. (2018a)), which composes of a sort pooling layer followed by a two-layer convolution network. In this experiment, we use the sort pooling layer to pull out the sorted features from the top 10 nodes (sorted by the largest value of each node).

For our method and SEAL, the prediction of link probability starts with a 32-length long graph embedding vector. After the pooling layer, we send the graph embedding to a dense layer with 32 neurons and another dense layer to generate the output. For GAE, the prediction of link probability requires the node level features of the two target nodes. In our implementation, after the GNN step, we simply look up the node embedding vectors for the two target nodes and concatenate the two vectors. The merged embedding is then sent to the same MLP as described earlier to generate the prediction.

In our methods, we use a three layer MLP with 32 neurons as the discriminator for the graph embedding. The input of the discriminator is the concatenation of two random graph embedding and the target of the discriminator is to predict whether these two embedding come from the same domain. In the actual implementation, because we train the model using a mini-batch of 32, this could be easily done by separating one batch of 32 embedding vectors to two groups of 16 vectors and comparing between these two groups.

In addition, we implement a 20 iteration warm-up stage for both our method and SEAL. In the warm-up stage, the models are trained without discriminator. The purpose of setting up the warm-up stage is to make it easier to train the adversarial model. As reported in literature, adversarial models are harder to train. If we train the model and the discriminator together from the very beginning, a bad initialization on the model may make it too difficult for the model to confuse the discriminator, which will lead to the "model collapsing". A 20 step warm-up stage may not always bring the model to the best spot but it offers a good starting point in most cases.

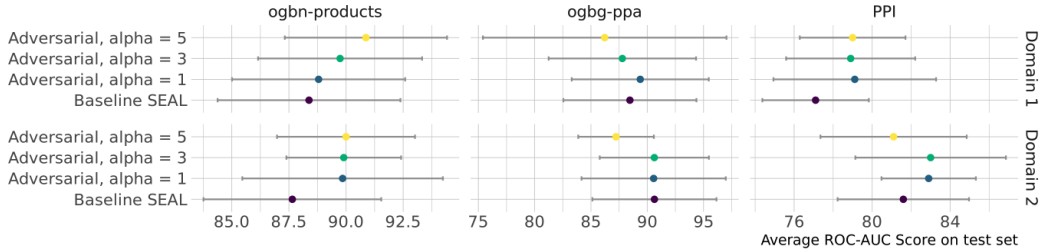

Figure 2: **Comparison of average test ROC-AUC performances in each dataset.** The colored dots represent the mean performance of 10 experiments and the error bars represent the standard deviation.

All the models are trained with a learning rate at 0.0001 with L2 regularization at 0.001 for 250 epochs. The discriminators are trained with a learning rate at 0.00001. All the experiments are trained and tested on a NVIDIA V100 card.

## 4.4 EXPERIMENT RESULTS

In our proposed method, $\alpha$ is an important hyperparameter because it controls the impact of the discriminator on the model. In this experiment, we select $\alpha$ from 1, 3, and 5 and we compare the results with all of our baseline methods. The results are displayed in Table 1 and Figure 2.

Table 1: Comparison of Test ROC-AUC Scores with Baseline Methods

|  | Adversarial Methods | | | SEAL | GAE | CN | AA | PPR |
|---|---|---|---|---|---|---|---|---|
|  | $\alpha = 1$ | $\alpha = 3$ | $\alpha = 5$ |  |  |  |  |  |
| ogbn-products |  |  |  |  |  |  |  |  |
| Domain 1 | 88.8±3.8 | 89.7±3.6 | **90.9±3.5** | 88.4±4.0 | 52.0±4.0 | 45.4 | 45.9 | 44.6 |
| Domain 2 | 89.8±4.4 | 89.9±2.5 | **90.0±3.0** | 87.7±3.9 | 58.4±4.0 | 33.4 | 30.1 | 43.6 |
| ogbg-ppa |  |  |  |  |  |  |  |  |
| Domain 1 | **89.4±6.1** | 87.8±6.6 | 86.2±10.8 | 88.5±5.9 | 52.7±6.8 | 39.2 | 39.1 | 38.2 |
| Domain 2 | **90.6±6.4** | 90.6±4.9 | 87.2±3.4 | 90.6±5.5 | 65.1±5.4 | 53.7 | 52.2 | 57.3 |
| PPI |  |  |  |  |  |  |  |  |
| Domain 1 | **79.1±4.2** | 78.9±3.3 | 79.0±2.7 | 77.1±2.7 | 55.0±3.2 | 57.0 | 63.0 | 43.5 |
| Domain 2 | 82.9±2.4 | **83.0±3.9** | 81.1±3.8 | 81.6±3.4 | 50.5±3.1 | 53.5 | 57.5 | 67.0 |

[1] Results from trainable NN models are averages (± standard deviation) of 10 experiments.

[2] Results from heuristic methods are direct output on the same test set.

First, we observe that compared with the vanilla SEAL method, in most cases, the proposed adversarial methods yield better performances over underrepresented domains. The average improvement is around 1.5% but the levels of improvement vary across different datasets. In the ogbn-products and PPI dataset, we observe much better performance from the adversarial models but in the ogbg-ppa dataset, the improvement is not very significant. We think the reason is that the feature distributions in the ogbg-ppa dataset are highly similar across domains. In this case, the impact of the lack of training data for one specific domain is very small. Therefore, the discriminator here cannot generate any benefits. We will discuss the details of our reasoning later.

The GAE method did not do well under this experiment. Unlike SEAL, GAE lacks the key node feature to describe the distance between each node to the target link. Under this few-shot learning setting, the limited number of nodes for training further restricts GAE's ability to efficiently train the model.

The performances of the heuristic methods are low as well. One possible explanation is that the link prediction task in these three experiment dataset heavily depends on node/edge features. Since all

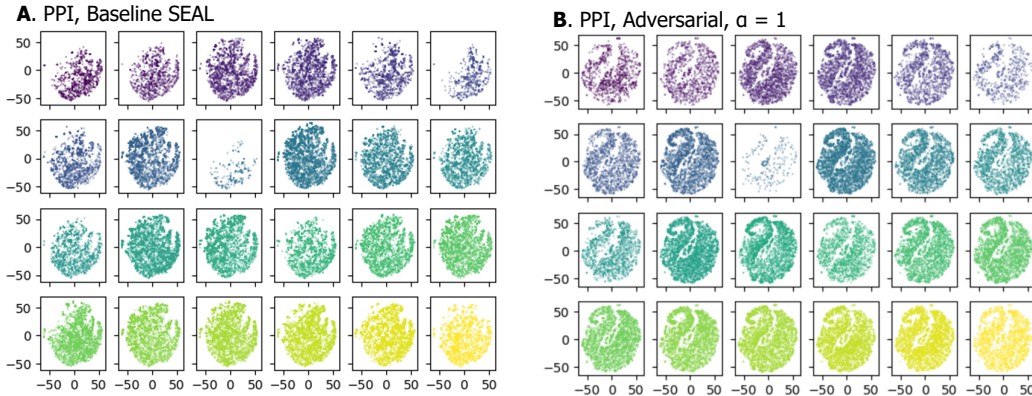

Figure 3: **Comparison of t-SNE visualization of graph embedding for all 24 domains in the PPI dataset.** The distribution of graph embeddings generated by baseline SEAL vary across domains while the graph embeddings generated by the adversarial method are domain-invariant.

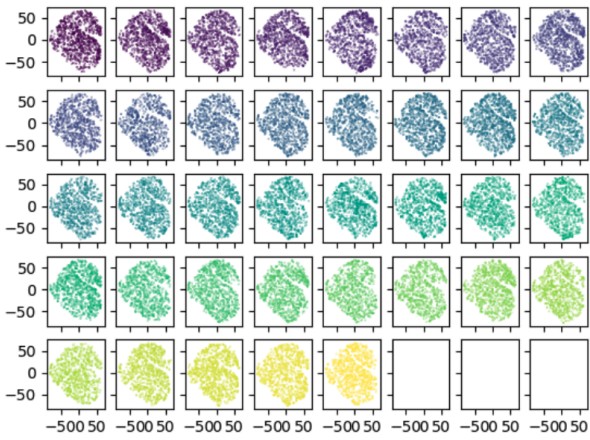

Figure 4: Domains in the ogbg-ppa dataset are already similar with baseline SEAL

three heuristic methods only use graph topological structure data, they lack enough information to make good predictions.

**Impact of Adversarial Training on Graph Embedding**

To understand the real impact of the adversarial training on graph embedding, we use t-SNE (van der Maaten & Hinton (2008)) to reduce the dimension of graph embedding and visualize them in 2-D space. Figure 3 shows the changes of graph embedding for all 24 domains in PPI. From the figure, we can see the dots generated by the adversarial method are more evenly distributed across the 2-D space and this observation further confirms that our proposed method is working as expected.

In addition, by inspecting the t-SNE plot (Figure 4) for the ogbg-ppa dataset, we find that with the baseline SEAL method, the distribution of embedding points among different domains in ogbg-ppa are already evenly distributed. It shows that there are few inter-domain differences in this dataset. Although the protein graphs come from different organisms, it could be the case that all the graphs are assembled in a similar way, so they are topologically similar. It could also be possible that the 7-dimension edge feature extracted for this dataset are not reflecting domain differences.

In fact, comparing Figure 3 and Figure 4 shows that the t-SNE visualization of the graph embedding vector is a good tool for machine learning practitioners to decide when to adopt the proposed

approach. If there are distinguishable domain differences across domains based on the t-SNE plot, the proposed method will likely yield some improvements.

**Training Curves, Run time and Impact of** $\alpha$

We also look into the impact of the adversarial design on the training curves. Figure 5 shows the smoothed training curves of 10 experiments with the ogbn-products dataset on domain 2. The shaded area represent the standard error. Note that we excluded the performances during the 20-step warm-up stage to zoom on the most important area. As presented in Figure 5, the baseline performance quickly drops after 50 iterations even with the use of L2 regularization. This is mostly due to the fact that the evaluation domain is underrepresented in training data (recall that in the training data, we have 1 shot of data in the evaluation domain and 10 shots of data for every other domain). By enforcing the graph embedding from different domains to be similar using adversarial training, the graph embedding generated by our method is more likely to work well with the evaluation domain. Therefore we see the performances from all three adversarial experiments reach better best performances compared with the baseline SEAL method.

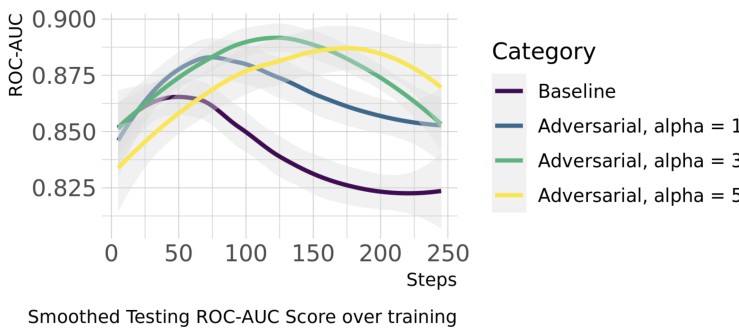

Figure 5: Comparison of Smoothed Training Curves over 10 Experiments

In addition, by increasing the value of $\alpha$, we increase the impact of the discriminator of the target model. In this case, $\alpha = 3$ seems to provide the best performance. However, if we look at the average best performances, the best $\alpha$ is in fact 5. A closer look into each individual training curve tells us that some of the best models were trained when $\alpha = 5$. However, the large $\alpha$ value also makes the model less stable and it's more likely to see the model collapse during the training process. Therefore, we conclude that when domain differences exist, higher impact of the discriminator helps the model reach better performances but it makes it more difficult to train. Eventually, there is a sweet spot to keep the balance between performance and model stability.

Figure 5 also tells us that compared with the baseline SEAL method, our method takes more steps to converge. Increasing the value of $\alpha$ increases the steps to converge as well. At the same time, it also takes more time to train the adversarial model because the computation of graph embedding needs to happen two times to collect gradients to train the target model and the discriminator. On average, it takes 1.134s to finish one training step for the ogbn-products dataset using baseline SEAL on an NVIDIA V100 graphic card, but it takes 1.406s (24.0 % increase) to train the network using adversarial method.

## 5    CONCLUSION

In this work, we propose an adversarial training-based approach to solve the few-shot graph link pre-diction problem. Compared with the state-of-arts method and other baselines, our method achieves better performance on underrepresented domains, especially when domain variance is not trivial. We further examine the changes of the graph embeddings using t-SNE and reveal the root of this improvement. At the same time, by showing the t-SNE visualizations for cases when the proposed method doesn't provide benefits, we show that t-SNE is a very efficient tool for practitioners to determine benefit of our method.

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
