# OpenReview forum: "Few-shot graph link prediction with domain adaptation"
_ICLR.cc/2022/Conference — ICLR 2022 Submitted_

### Official Review · Reviewer_1WRx · 2021-10-31

**Correctness:** 3
**Technical Novelty And Significance:** 2
**Empirical Novelty And Significance:** 2
**Recommendation:** 3
**Confidence:** 5

**Main Review:**

Strength:

This paper introduces adversarial learning (which is widely used in UDA methods) to few-shot graph link prediction problems to learn domain-invariant representations.

Weakness:
1.	The motivation of this work should be further justified. In few-shot learning, we usually consider how to leverage a few instances to learn a generalizable model. This paper defines and creates a few-shot situation for graph link prediction, but the proposed method does not consider how to effectively use “few-shot” and how to guarantee the trained model can be generalized well to new tasks with 0/few training steps.
2.	The definition of “domain” in this paper is unclear. For instance, why select multiple domains from the same single graph in ogbn-products? Should we consider the selected domains as “different domains”?
3.	The application of adversarial learning in few-shot learning is confusing. Adversarial learning in domain adaptation aims to learn domain-invariant representations, but why do we need such kind of representation in few-shot learning?


**Summary Of The Paper:**

This paper proposes a method for few-shot graph link prediction by using a domain discriminator. The motivation is to learn domain-invariant graph-level representations. The authors also introduce the concept of “shot” in the graph. The proposed method outperforms baselines on three different datasets.

**Summary Of The Review:**

This paper is not well motivated and needs further justification.

---

> ### Author Response · Authors · 2021-11-23
> **Response to Reviewer #4**
>
> Dear Reviewer,
>
> Thank you so much for the critical and insightful comments. After reading your suggestions, we realized that some of our previous statements about our motivation may have caused some confusion. We have completely rewritten our introduction section on the first page and several other parts of the paper to address that. We hope it will clarify the motivation, setting, and experimental design of our work.
>
> We next detail how we address each of your comments.
>
> **1. The motivation of this work should be further justified. In few-shot learning, we usually consider how to leverage a few instances to learn a generalizable model. This paper defines and creates a few-shot situation for graph link prediction, but the proposed method does not consider how to effectively use “few-shot” and how to guarantee the trained model can be generalized well to new tasks with 0/few training steps.**
>
> We have completely updated our introduction to explain our motivation and our setting better. We could not carry out additional experiments to test our model on novel domains to validate our framework's generalizability for new tasks.
>
> Previous works in CV (Motiian et al. (2017), Ganin et al. (2016)) have already shown that creating domain agnostic features helps reduce the number of samples needed for each class. And therefore create more robust models that can work with domains with few-shot or even novel domains.
>
> **2. The definition of “domain” in this paper is unclear. For instance, why select multiple domains from the same single graph in ogbn-products? Should we consider the selected domains as “different domains”?**
>
> Yes, in the updated document, we provide examples and descriptions on the meaning of domain in the first introduction paragraph on page 1. We also provide a more precise mathematical definition of “domain” in section 3.3 on page 4. Regarding your question here, we selected multiple domains to simulate the imbalanced data problem, as we described in the first paragraph. We would consider the selected domain with one shot as our target domain and the rest as source domains.
>
> >**1 Introduction** Real-world link prediction problems often deal with graphs with nodes from various domains. For example, we can separate the items from a product purchase network into multiple domains such as books, electronics, luxury items. A financial transaction network can have data coming from various countries. Different domains may have different distributions, but more importantly, domains are often not balanced in most real-world datasets. For example, we may expect to see hundreds of thousands of nodes for books but only a few hundred for luxury items. There might also be billions of financial transactions in developed countries but much fewer in developing countries. We observe a similar phenomenon even in academic graph datasets. For example, in the ogbn-arxiv (Hu et al.(2020)) dataset, which includes pre-prints from 40 computer science subjects, papers from the most popular subject takes up to 16.13% of all the nodes. In contrast, the least represented subject only has 0.02% of the data.
>
> >**3.3 Domains** For our setting, we have a total of $m$ domains and each domain has one or more graphs or subgraphs $D^k = \\{G_1^k,...\\}_{k=1}^m$, where $D^k$ denotes the $k^{th}$ domain and graph $G_1^k$ is a realization of random variable $\mathcal{G}^k$. The effective of our method compared to baseline is based on the assumption that the distribution of the random variable $\mathcal{G}$ is different across different domains $p(\mathcal{G}^k) \neq p(\mathcal{G}^{k'})$ if $k' \neq k$.
>
> **3. The application of adversarial learning in few-shot learning is confusing. Adversarial learning in domain adaptation aims to learn domain-invariant representations, but why do we need such kind of representation in few-shot learning?**
>
> In the updated document, we provided an intuition explaining why we think learning domain-invariant representation is helpful for few-shot learning in the third introduction paragraph on page 1. The CV literature has already explored the idea of using adversarial learning to build domain-agnostic features, and people have found that doing so could reduce the number of training samples needed for each class. You can check the first paragraph on Page 2 for details.
>
> >(We are running out of space so please refer to the updated document for original text)
>
> Please let us know if you have additional comments, and we will be happy to address them. Thanks a lot for reconsidering this paper!
>
> Best,
>
> Authors

---

> > ### Comment · Reviewer_1WRx · 2021-11-30
> > **Thanks for the response**
> >
> > I appreciate the authors’ efforts in answering my questions. However, I am still not convinced of the motivation of this work. In addition, the experimental setting cannot prove the applicability and effectiveness in more complex scenarios, which is also pointed by reviewer XgHj. I keep my score.

---

### Official Review · Reviewer_XgHj · 2021-11-02

**Correctness:** 3
**Technical Novelty And Significance:** 2
**Empirical Novelty And Significance:** 2
**Recommendation:** 5
**Confidence:** 4

**Main Review:**

Strength
The proposed domain adaptive method of using adversarial training is well described.
The experimental results support the effectiveness of the proposed method.
The T-SNE plots add good insights of the proposed method, can facilitate its applicability.

Weakness
The paper hasn’t discussed the key difference between the proposed method and the DANN (Domain-Adversarial NN). It made it harder to assess the novelty of the proposed method. The paper seems like a direct application of DANN on a new problem, link prediction using GNN.

The paper’s experimental settings might not be close to a real world scenario. The paper selects 10 samples for other domains and only 1 sample for an imbalanced domain. How likely is this a real world scenario? This simple scenario makes it hard to justify the universal applicability of the proposed method.


**Summary Of The Paper:**

The paper proposes a method to resolve the issue of domain imbalanced dataset in graph link prediction. The proposed method uses adversarial training to generate graph embeddings that are domain agnostic in order to facilitate transfer learning cross domains. The paper uses T-SNE plot of graph embedding to gain insights of the best scenarios for applying the proposed methods. The paper compares the proposed method with heuristics-based and GNN-based domain adaptation methods using experiments.


**Summary Of The Review:**

The paper should clearly claim its novelty and run more experiments that are closer to real-world scenarios. I am going with a rating to marginally reject the paper, but willing to change the rating if authors can address the above concerns.

---

> ### Author Response · Authors · 2021-11-23
> **Response to Reviewer #3**
>
> Dear Reviewer,
>
> Thank you so much for the critical and insightful comments. After reading your suggestions, we realized that some of our previous statements about our motivation may have caused some confusion. We have completely rewritten our introduction section on the first page and several other parts of the paper to address that. We hope it will clarify the motivation, setting, and experimental design of our work.
>
> We next detail how we address each of your comments.
>
> **1. The paper hasn’t discussed the key difference between the proposed method and the DANN (Domain-Adversarial NN). It made it harder to assess the novelty of the proposed method. The paper seems like a direct application of DANN on a new problem, link prediction using GNN.**
>
> Yes, our method follows the track of DANN to build domain agnostic features, but a key idea is to further improve it with the enclosing subgraph idea proposed in SEAL. Please check the top two paragraphs on page 2 as cited below for clarification on it.
>
> >In addition, the CV literature has explored the idea of using adversarial learning to build better domain agnostic features. Domain-Adversarial Neural Network (DANN) (Ganin et al. (2016)) first introduces the idea of using a discriminator to align and separate semantic probability distributions. Further research (Motiian et al. (2017)) shows that it helps reduce the number of samples needed for each domain/class. Essentially, our work follows this path but further improves it with the enclosing subgraph idea to achieve better performances for graph link predictions.
>
> >We find that the method of extracting enclosing subgraphs, proposed in the SEAL model Zhang & Chen (2018) suits well for our setting because it not only boosts the number of trainable samples but also gives them different topological features. It is similar to what ``cropping'' does for images. CV Research has shown that data augmentation method, such as cropping  (Qi et al. (2018); Zhang et al.(2018b)), introduces invariances for models to capture and is effective for FSL problems.
>
> **2. The paper’s experimental settings might not be close to a real world scenario. The paper selects 10 samples for other domains and only 1 sample for an imbalanced domain. How likely is this a real world scenario? This simple scenario makes it hard to justify the universal applicability of the proposed method.**
>
> We have updated the introduction with a few examples and supporting data. We think it is very common to see a highly skewed real-world dataset with domains with very few data points. For example, in the ogbn-arxiv benchmark dataset, the most popular domain is more than 100 times bigger than the least abundant domain. In this work, we carry out a controlled experiment that simulates this scenario.
>
> >**1 Introduction** Real-world link prediction problems often deal with graphs with nodes from various domains. For example, we can separate the items from a product purchase network into multiple domains such as books, electronics, luxury items. A financial transaction network can have data coming from various countries. Different domains may have different distributions, but more importantly, domains are often not balanced in most real-world datasets. For example, we may expect to see hundreds of thousands of nodes for books but only a few hundred for luxury items. There might also be billions of financial transactions in developed countries but much fewer in developing countries. We observe a similar phenomenon even in academic graph datasets. For example, in the ogbn-arxiv (Hu et al.(2020)) dataset, which includes pre-prints from 40 computer science subjects, papers from the most popular subject takes up to 16.13% of all the nodes. In contrast, the least represented subject only has 0.02% of the data.
>
> Please let us know if you have additional comments, and we will be happy to address them. Thanks a lot for reconsidering this paper!
>
> Best,
>
> Authors

---

> > ### Comment · Reviewer_XgHj · 2021-11-30
> > **Response**
> >
> > I appreciate that the authors took time to answer the questions and made corresponding revisions to the paper. I still believe that the claim of novelty of using enclosing subgraph is minor and there is no additional experiments on more realistic datasets. I have decided to keep my initial rating.

---

### Official Review · Reviewer_Vi6R · 2021-11-02

**Correctness:** 4
**Technical Novelty And Significance:** 4
**Empirical Novelty And Significance:** 4
**Recommendation:** 8
**Confidence:** 5

**Main Review:**

Strengths: Application of few shot learning methodology to predicting graph links is novel. Will have applications beyond the benchmark datasets used in the paper.
Weaknesses: I like this paper and so do not see any obvious weaknesses.

**Summary Of The Paper:**

This paper presents a novel approach to  link prediction methods in graph representation of imbalanced domains using adversarial training. the result is an improved domain-agnostic graph embeddings. The approach takes is similar to few-shot learning that is popular in computer vision. Discussions on how to define shots for graphs, and how to design experiments for addressing imbalanced domains contribute to the novelty of the paper. Results on two Stanford Open Graph Benchmarks and the PPI dataset are given.


**Summary Of The Review:**

A novel application of few shot learning and domain adaptation to machine learning from graph representations.

---

> ### Author Response · Authors · 2021-11-23
> **Response to Reviewer #2**
>
> Dear Reviewer,
>
> We are truly thankful for your review! Thank you so much for your kind support and encouragement!
>
> Best,
>
> Authors

---

### Official Review · Reviewer_YcHW · 2021-11-04

**Correctness:** 3
**Technical Novelty And Significance:** 2
**Empirical Novelty And Significance:** 2
**Recommendation:** 5
**Confidence:** 3

**Main Review:**

#### Strength

- The paper first proposed a few-shot link problem with imbalanced domains.
- The proposed method achieves good performance compared to baselines and the authors use t-SNE plots as a rule of thumb for practitioners to decide whether to incorporate the proposed method.
- Visual explanations of the source of improvements also validate the proposed approach.

#### Weakness

- Use adversarial training in graph or link prediction is not novel. There is a lot of prior work which is not discussed in the paper.

[1] Lei, K., Qin, M., Bai, B., Zhang, G., and Yang, M., 2019, April. Gcn-gan: A non-linear temporal link prediction model for weighted dynamic networks. In IEEE INFOCOM 2019-IEEE Conference on Computer Communications (pp. 388-396). IEEE.
[2] Wang, H., Wang, J., Wang, J., Zhao, M., Zhang, W., Zhang, F., Xie, X. and Guo, M., 2018, April. Graphgan: Graph representation learning with generative adversarial nets. In Proceedings of the AAAI conference on artificial intelligence (Vol. 32, No. 1).

- The proposed few-shot link prediction setting is kind of arbitrary, what is the application and motivation behind this setting? why the last domain has to be 1 shot?

- The model is a three layers GCN with 32 neurons. there are no ablations for the hyper-parameters for the model.

**Summary Of The Paper:**

This paper proposed adversarial training for few-shot link prediction problems. The authors introduce a domain discriminator on pairs
of graph-level embedding and address the issue of few-shot learning in graph data. The proposed method is tested on 3 benchmark datasets and achieves good performance compared to the prior approaches.

**Summary Of The Review:**

The paper proposed an interesting few-shot link prediction problem. However, important related work is not discussed in the paper. More work should stress the difference between the proposed method and prior work.

---

> ### Author Response · Authors · 2021-11-23
> **Response to Reviewer #1**
>
> Dear Reviewer,
>
> Thank you so much for the critical and insightful comments. After reading your suggestions, we realized that some of our previous statements about our motivation may have caused some confusion. We have completely rewritten our introduction section on the first page and several other parts of the paper to address that. We hope it will clarify the motivation, setting, and experimental design of our work.
>
> We next detail how we address each of your comments.
>
> **1. Use adversarial training in graph or link prediction is not novel. There is a lot of prior work which is not discussed in the paper.**
>
> Yes, we agree that the use of adversarial training in link prediction on a graph is not novel. Our setting is a bit different; we use adversarial training to enforce the model to learn domain agnostic representations, which aid in adapting the models for sparse domains. We have added a new section (section 2.3 on page 3) to clarify this issue.
>
> >**2.3 Adversarial Learning** To learn domain agnostic models, we need to ensure that the learned graph embedding does not contain domain-specific information. To achieve this, a commonly used technique is adversarial learning, in which two models are trained against each other and play a minimax game based on game theory. For example, in the popular Generative Adversarial Network (GAN) (Goodfellow et al.(2014)), adversarial learning helps generators build more realistic images by having a discriminator that identifies whether an image is synthetic or real. There have been prior works done on using adversarial training on graph-related tasks. The work from Wang et al. (2018) uses GAN to improve various graph-related tasks, such as link prediction and node property prediction, by generating ``fake'' graphs. The work from Lei et al. (2019) also uses GAN to solve the challenging temporal link prediction problem for weighted dynamic networks.
>
> **2. The proposed few-shot link prediction setting is kind of arbitrary, what is the application and motivation behind this setting? why the last domain has to be 1 shot?**
>
> We have rewritten our introduction to clarify the motivation of our problem. Setting the last domain to a single shot is part of our controlled experiments to simulate the distribution of sparse domains. In real-world datasets, data across domains are even more skewed than our experimental setup.
>
> >**1 Introduction** Real-world link prediction problems often deal with graphs with nodes from various domains. For example, we can separate the items from a product purchase network into multiple domains such as books, electronics, luxury items. A financial transaction network can have data coming from various countries. Different domains may have different distributions, but more importantly, domains are often not balanced in most real-world datasets. For example, we may expect to see hundreds of thousands of nodes for books but only a few hundred for luxury items. There might also be billions of financial transactions in developed countries but much fewer in developing countries. We observe a similar phenomenon even in academic graph datasets. For example, in the ogbn-arxiv (Hu et al.(2020)) dataset, which includes pre-prints from 40 computer science subjects, papers from the most popular subject takes up to 16.13\% of all the nodes. In contrast, the least represented subject only has 0.02\% of the data.
>
> **3. The model is a three layers GCN with 32 neurons. there are no ablations for the hyper-parameters for the model.**
>
> Although we didn’t state it clearly in the paper, we did a hyperparameter search using a bare-bone link prediction model on one dataset. For the other two datasets, they are sub-optimal. For simplicity of paper writing, we chose to use a flat architecture for all three datasets. After all, the structure of GNN is not the primary purpose of this paper.
>
> Please let us know if you have additional comments, and we will be happy to address them. Thanks a lot for reconsidering this paper!
>
> Best,
>
> Authors

---

### Decision · Program_Chairs · 2022-01-20

**Decision:**

Reject

**Comment:**

This paper has been reviewed by four expert reviewers who gave diverging scores. The three negative reviewers have provided significant constructive feedback. The main criticism is the lack of novelty and clarity in the paper. The authors have submitted their rebuttal which did not improve the scores of these reviewers. After the discussion phase, the paper did not obtain any support for acceptance and stayed under the acceptance threshold. Following the reviewers' recommendation, the meta reviewer recommends rejection.